# Phase Behavior of Aqueous Mixtures of Sodium Alginate with Fish Gelatin: Effects of pH and Ionic Strength

**DOI:** 10.3390/polym15102253

**Published:** 2023-05-10

**Authors:** Daria S. Kolotova, Ekaterina V. Borovinskaya, Vlada V. Bordiyan, Yuriy F. Zuev, Vadim V. Salnikov, Olga S. Zueva, Svetlana R. Derkach

**Affiliations:** 1Laboratory of Chemistry and Technology of Marine Bioresources, Institute of Natural Science and Technology, Murmansk State Technical University, Murmansk 183010, Russia; shibekoev2@mstu.edu.ru (E.V.B.); bordiyanvv@mstu.edu.ru (V.V.B.); 2Kazan Institute of Biochemistry and Biophysics, FRC Kazan Scientific Center of RAS, Kazan 420111, Russia; yufzuev@mail.ru (Y.F.Z.); vadim.salnikov.56@mail.ru (V.V.S.); 3A. Butlerov Chemical Institute, Kazan Federal University, Kazan 420008, Russia; 4Institute of Electric Power Engineering and Electronics, Kazan State Power Engineering University, Kazan 420066, Russia; ostefzueva@mail.ru

**Keywords:** fish gelatin, sodium alginate, polyelectrolyte complexes, complex coacervation, effect of pH, ionic strength

## Abstract

The phase behavior of aqueous mixtures of fish gelatin (FG) and sodium alginate (SA) and complex coacervation phenomena depending on pH, ionic strength, and cation type (Na^+^, Ca^2+^) were studied by turbidimetric acid titration, UV spectrophotometry, dynamic light scattering, transmission electron microscopy and scanning electron microscopy for different mass ratios of sodium alginate and gelatin (Z = 0.01–1.00). The boundary pH values determining the formation and dissociation of SA-FG complexes were measured, and we found that the formation of soluble SA-FG complexes occurs in the transition from neutral (pH_c_) to acidic (pH_φ1_) conditions. Insoluble complexes formed below pH_φ1_ separate into distinct phases, and the phenomenon of complex coacervation is thus observed. Formation of the highest number of insoluble SA-FG complexes, based on the value of the absorption maximum, is observed at рH_opt_ and results from strong electrostatic interactions. Then, visible aggregation occurs, and dissociation of the complexes is observed when the next boundary, pH_φ2_, is reached. As Z increases in the range of SA-FG mass ratios from 0.01 to 1.00, the boundary values of рН_c_, рH_φ1_, рH_opt_, and рH_φ2_ become more acidic, shifting from 7.0 to 4.6, from 6.8 to 4.3, from 6.6 to 2.8, and from 6.0 to 2.7, respectively. An increase in ionic strength leads to suppression of the electrostatic interaction between the FG and SA molecules, and no complex coacervation is observed at NaCl and CaCl_2_ concentrations of 50 to 200 mM.

## 1. Introduction

Interactions between proteins and polysaccharides have been actively studied in previous decades due to the widespread application of these compounds in food technologies, the cosmetic industry, medicine, pharmaceutics, tissue engineering, and many other fields [1,2,3,4,5,6,7,8]. In particular, proteins are used as emulsifying, foaming, gelling, or structuring agents in the food industry to develop liquid or solid matrices of functional food products [9] and in drug delivery systems [10] and for the creation of films, coatings, and capsules in the food, pharmaceutical, and cosmetic industries [11,12,13]. Polysaccharides are also widely used in many industries as thickeners or moisture-retaining, gelling, drying, or structuring agents [14,15,16].

Depending on internal and external factors, the interaction between proteins and polysaccharides in aqueous media can adopt two directions: either attraction or repulsion [17,18]. Such factors include the molecular characteristics of proteins and polysaccharides (molecular weight, conformation, chain flexibility, type and number of reaction groups, chain charge density); solvent properties (pH, ionic strength, polarity); and mixing conditions (temperature, time, pressure, ratio of protein and polysaccharide, total concentration of biopolymers, presence of crosslinking agents) [17,19,20,21].

Attractive interactions between proteins and polysaccharides occur mainly due to electrostatic interactions between positively charged protein groups and negatively charged polysaccharide groups [19,21,22]. Such interactions lead to the formation of soluble or insoluble protein–polysaccharide complexes. The insoluble complexes then separate, forming a two-phase system consisting of the phases containing the complexes and the solvent. This phase separation phenomenon is also called “complex coacervation” if the separated insoluble complexes are in the liquid state or “precipitation” if the complexes are in the solid state.

It is known that flexible weakly charged anionic polysaccharides, such as gum arabic, hyaluronic acid, dextran sulfate, and some pectins, tend to form liquid coacervates with positively charged proteins due to their relatively weak electrostatic interaction [21,23]. On the other hand, rigid, strongly charged anionic polysaccharides, such as gellan gum, sodium alginate, and κ-carrageenan, tend to form solid coacervates due to their stronger electrostatic attraction with proteins [17,21].

Repulsive interactions are caused by the thermodynamic incompatibility of equally charged or uncharged biopolymers [24,25].

Controlling protein–polysaccharide interactions could be a promising strategy for overcoming the disadvantages when using each biopolymer separately and improving their properties. In addition, the polyelectrolyte complex formation method is popular due to its low cost, low energy consumption, and efficiency compared with conventional processes such as solvent evaporation, emulsification, polymerization, etc. [26,27]. The formation of such non-covalent complexes does not require the use of organic solvents or crosslinking agents [28,29].

Many works have reported the interaction between proteins and polysaccharides—which are simultaneously biopolymers and polyelectrolytes—such as gelatin–hummiarabic [30], soy protein isolate–pectin [31,32], canola protein isolate–(θ-, η- and ι-) carrageenan [33], and others. Protein–polysaccharide interactions, including the formation of polyelectrolyte complexes, have been discussed in detail in previous reviews [6,21,34]. Water-soluble polyelectrolyte complexes are formed only under strictly defined conditions (ratio of components, pH) [35,36].

Systems containing gelatin and polysaccharides of marine origin are among the most promising systems currently used in biomedicine, pharmaceuticals, and the food industry due to the high availability, low cost, safety, and biodegradability of these materials [37,38,39]. Gelatin, being a polyampholyte, can form complexes with both cationic and anionic polysaccharides depending on the pH of the medium. In the literature, there are many papers devoted to studying the interactions of gelatin with polysaccharides of different origins, such as gum arabic [22], sodium alginate [40], and κ-carrageenan [41]. The process of gene delivery using polyelectrolyte nanoparticles derived from cationized gelatin and anionic polysaccharides, dextran sulfate, and chondroitin sulfate was studied in [42]. Despite the fact that there are many research articles demonstrating the potential application of polyelectrolyte polysaccharide–protein complexes in drug delivery, the detailed characteristics of these materials have not yet been described. From a practical point of view, the establishment of a correlation between the structures of such complexes and their properties is particularly interesting.

Thus, the purpose of this investigation is to study the phase behavior of aqueous mixtures of sodium alginate with fish gelatin during complexation. This includes assessing the effects of pH and ionic strength on the phase behavior of SA-FG aqueous mixtures.

## 2. Materials and Methods

### 2.1. Materials

Sodium alginate from brown algae (A2033, Sigma, Gillingham, UK) was used. The average mass molecular weight of sodium alginate was determined by HPLC using an LC-20A chromatograph with a RID-10A refractive index detector (Shimadzu Corp., Kyoto, Japan) with a Shodex Asahipak GS-520 HQ and GS-620 HQ (7.5 mm × 300 mm). The molecular mass distribution of alginate was assessed by normalizing the peak areas [43]. The average mass molecular weight (M_w_) of sodium alginate is ~507 kDa. The amount of alginic acid was determined on the basis of a color reaction with 3,5-dimethylphenol, and sulfuric acid was used as a reference [44]. The alginic acid content of the sodium alginate sample was (92.2 ± 0.7)%.

Fish gelatin from Atlantic Cod skin was obtained according to the method described in [45] for use in experiments. After being preliminarily cleaned from scales and residual muscle tissue, the cod skin was thawed, chopped into 5 mm × 5 mm pieces, and then treated with ethanol to remove fat. Next, the cod skin was mixed with distilled water at a ratio of 1:3 (by wt%) and stirred for 10 min. Gelatin extraction was carried out at pH = 5 for 3 h at 50 ± 1 °C with constant stirring at a speed of 80–100 rpm. The medium pH was adjusted by adding glacial acetic acid. After extraction, the reaction mixture was neutralized to pH 5.5–6.0 and then filtered by vacuum filtration at 30 °C through a paper filter with a pore diameter of 12 µm. The resulting filtrate (gelatin aqueous dispersion) was dried in a FreeZone lyophilic dryer (Labconco, USA) at −50 °C and a residual pressure of 2.4–2.6 Pa. The molecular weight of fish gelatin was determined to be 153 kDa by HPLC at a wavelength of 280 nm using the LCMS-QP8000 chromatograph (Shimadzu, Japan). The isoelectric point of gelatin (pI) was measured as 7.4 with the turbidimetric method and as 7.1 with the viscosimetric method. The amino acid composition of the gelatin is given in Table 1.

### 2.2. Aqueous Mixtures of Sodium Alginate with Fish Gelatin

Aqueous dispersions of fish gelatin (FG) and sodium alginate (SA) with a concentration of 0.2 wt% were separately prepared by dissolution in distilled water overnight at 25 °C and constant stirring. Then, aqueous mixtures of alginate with gelatin were prepared by adding the SA dispersion to the FG. The mass ratio of sodium alginate and gelatin (Z = C_SA_/C_FG_, g_SA_/g*_F_*_G_) in the aqueous mixtures varied between 0.01 and 1.00 with a constant concentration of gelatin of C_FG_ = 0.1 wt%. To study the effect of pH on the formation of *SA-FG* complexes, the pH of the mixtures was preliminarily adjusted to ~8 using 0.1 M NaOH. The effect of the ionic strength on the interaction of FG with SA was studied by adding sodium and calcium chlorides (1–200 mM) at Z = 0.07 and C_FG_ = 0.1 wt%.

### 2.3. Turbidimetric and UV Absorption Spectrum Measurements

Ultraviolet spectral measurements of pure fish gelatin, sodium alginate, and aqueous mixtures of sodium alginate and fish gelatin (SA-FG) in the wavelength range of 190 to 700 nm were recorded at 25 °C with an accuracy of 1 nm using a UV–Vis spectrophotometer Т70 (PG Instruments, Wibtoft, United Kingdom). The width of the quartz cuvette was 1 cm. The deionized water was used as a blank sample. The concentration of biopolymers in individual dispersions was 0.1 wt%. The mass ratio of SA and FG in the mixtures ranged from 0.01 to 1.00, with a constant concentration of gelatin of C_FG_ = 0.1 wt%.

The interaction of FG with SA was investigated using the method of turbidimetric titration. The gelatin dispersion (C_FG_ = 0.2 wt%) was titrated with a dispersion of sodium alginate (C_SA_ = 0.2 wt%). The optical density was measured at a wavelength of λ = 400 nm and an optical path length of l = 1 cm using a spectrophotometer Т70 (PG Instruments, Wibtoft, United Kingdom). Measurements were carried out at room temperature (25 °C).

### 2.4. Turbidimetric Acid Titration

The turbidimetric acid titration method was used to study the phase behavior of aqueous mixtures of FG with SA at 25 °С. To study the effect of pH on the phase behavior of SA-FG mixtures with Z from 0.01 to 1, SA-FG aqueous mixtures were titrated to pH ~2 using acetic acid solutions of different concentrations (1–100%) to minimize dilution effects. During titration, the pH and optical density values were recorded. The optical density and pH of the mixtures were determined during titration at a wavelength of λ = 400 nm and an optical path length of l = 1 cm using a UV–Vis spectrophotometer T70 (PG Instruments, Wibtoft, United Kingdom) at 25 °C.

The critical pH values (pH_c_, pH_φ1_, pH_opt_, pH_φ2_) were determined from the dependence of absorbance on pH obtained during acid titration in accordance with the method described in [17,18,30], where pH_c_ is the limiting pH value at which there is initial weak interaction between the polymers with the formation of soluble complexes; pH_φ1_ is the limiting pH value below which the mixture becomes turbid due to the formation of insoluble sodium alginate–gelatin complexes; pH_φ2_ is the limiting pH value below which turbidity disappears due to dissociation of those complexes [18]; and pH_opt_ is the pH value at which the highest optical density was observed (Figure 1).

### 2.5. Particle Size Measurements

The average hydrodynamic radius of particles was measured by dynamic light scattering using the Photocor Complex-ZI analyzer (Photocor, Moscow, Russia). A thermally stabilized semiconductor laser (λ = 638 nm, 30 mW) was used as a light source. The samples were held for 1 h before measurements. Measurements of the SA-FG complexes’ hydrodynamic radii were carried out at a scattering angle of 90° and temperature of 25 °C. All of the measurements were done at least in triplicate.

### 2.6. Zeta Potential Measurements

Zeta potential was measured using an analyzer—Photocor Complex-ZI (Photocor, Moscow, Russia)—equipped with a thermally stabilized semiconductor laser (λ = 638 nm, 30 mW) as the light source. Doppler signal analysis was performed in a mode of phase analysis light scattering (PALS). Electrophoretic mobility μE of particles was converted to zeta potential using the Smoluchowski equation. All values of zeta potential were obtained at a scattering angle of 20°. The measurements were performed at 25 °C, and the samples were held for 60 min before the measurements. All of the measurements were done at least in triplicate.

### 2.7. Scanning Electron Microscopy (SEM) and Transmission Electron Microscopy (TEM)

Field emission scanning electron microscopy multipurpose analytical complex Merlin (Carl Zeiss, Oberkochen, Germany) and an HT7700 Exalens transmission electron microscope (Hitachi, Tokyo, Japan) were used to capture images of SA-FG mixtures.

## 3. Results

### 3.1. Influence of the SA-FG Mass Ratio

The UV absorption spectra of the SA, FG dispersions, and SA-FG aqueous mixtures were obtained at a mass ratio of biopolymers, Z (Z = C_SA_/C_FG_, g_SA_/g_FG_), from 0.01 to 0.1 (Figure 2). In sodium alginate aqueous dispersion, the absorption maximum was observed at a wavelength of 197 nm. This is due to the presence of hydroxyl and carboxyl groups in the molecule, which absorbs in the ultraviolet range [46]. For the fish gelatin dispersion, a broad absorption band was detected at 224 nm. Undivided nitrogen electron pairs conjugated to double bonds in histidine and arginine residues [46], as well as conjugated double bonds in the benzene ring of aromatic amino acids, especially tyrosine [47], were found to contribute significantly to the absorption band position of gelatin.

It was found that the introduction of sodium alginate into the gelatin dispersion led to a shift in the maximum absorption wavelength from 220 to 229 nm, and this was accompanied by an increase in absorbance and a significant broadening of the obtained absorption band (Figure 2). The observed changes in the spectra of gelatin are associated with electrostatic interactions between charged carboxyl groups of the β-D-mannuronic and α-L-guluronic acid residues of sodium alginate and the amino groups of gelatin. These changes were also observed as a result of the formation of hydrogen bonds between hydroxyl groups of sodium alginate and tyrosine residues of gelatin. The increase in the absorption intensity in this region of the spectrum is associated with light scattering by the particles of polyelectrolyte complexes.

The dependence of absorbance on the mass ratio of sodium alginate and fish gelatin was obtained (Figure 3). An increase in absorbance up to Z ≤ 0.07 is associated with an increase in the content of stoichiometric insoluble SA-FG complexes. During the formation of stoichiometric complexes, negatively charged sodium alginate molecules are completely masked by positively charged gelatin. These complexes exist in the system together with unbound macromolecules of gelatin. The observed maximum value at Z = 0.07 on the curve corresponds to the formation of the maximum observed number of stoichiometric complexes as a result of the mutual neutralization of charges and the most effective formation of ionic pairs. A further increase in Z led to a decrease in the optical density of mixtures due to the formation of soluble, nonstoichiometric sodium alginate–gelatin complexes of variable composition. With an increase in the content of sodium alginate, its negative charge becomes uncompensated, which leads to an increase in complex dissolution and a decrease in absorbance.

The zeta potential of SA-FG complexes was measured to better understand the effects of the SA-FG mass ratio on the electrostatic interactions between FG and SA. The zeta potential of the mixtures was between 9.06 and −4.47 mV in the range of SA-FG mass ratio values from 0.01 to 0.10 (Figure 4a). A decrease in the zeta potential, whose value was generally between the values of pure FG (9.24 mV) and SA (−14.66 mV), with an increase in Z indicates the charge compensation between FG and AL molecules. However, the value of the zeta potential did not approach zero at Z = 0.07; therefore, complete neutralization of charges at this value is not observed. This may indicate that the maximum absorbance observed in Figure 3 is associated with an increase in the electrostatic interaction between SA and FG, which led to the formation of larger light-scattering aggregates. Indeed, as can be seen from Figure 4b, the maximum value of the average hydrodynamic radius (379 nm) is observed at Z = 0.07.

### 3.2. Influence of рН

The phase behavior of SA-FG aqueous mixtures depends largely on the pH of the medium, the value of which affects, in turn, the degree of ionization of the biopolymers [17,22,23]. Figure 5a shows the UV spectra obtained for SA-FG aqueous mixtures at different pH values and Z = 0.07, at which the maximum value of optical density was observed. It can be seen that there is practically no shift in the absorption maximum in the pH range of 6 to 7. In this pH range, gelatin is predominantly negatively charged, although it carries a positive charge, and aqueous mixtures of SA-FG are transparent single-phase systems that contain individual molecules of biopolymers and their soluble complexes, which are formed due to the weak electrostatic interaction between the positive charge areas of the gelatin molecule and the negatively charged groups of sodium alginate molecules. However, the electrostatic repulsion between the molecular chains is greater than the electrostatic interaction. When the pH is decreased from 5 to 2, there is a significant shift in the absorption maximum from 212 to 237 nm (Figure 5b). At these pH values, gelatin is predominantly positively charged; hence, there is stronger electrostatic interaction between gelatin and sodium alginate. As a result, insoluble complexes are formed, which separate into distinct phases, and complex coacervation is thus observed.

The zeta potential of the SA-FG aqueous mixtures increased with decreasing pH from 8 to 2 due to the charge compensation between SA and FG molecules (Figure 5c). The SA-FG complexes are recharged in the pH range of 6 to 5, as evidenced by the change in the zeta potential from −4.39 to 2.30 mV, respectively. Consequently, in this pH range, there is almost complete neutralization of charges and the formation of insoluble complexes.

Figure 6 shows the dependence of the optical density of aqueous mixtures of gelatin and sodium alginate on pH obtained during acid titration. It can be seen that with an increasing mass ratio, Z, i.e., an increase in the content of sodium alginate in the system, the maximal optical density shifts to the acidic region of pH, and the system turbidity also increases. The increase in absorbance is connected with the formation of insoluble SA-FG complexes. At a pH corresponding to the curve maxima, the highest number of insoluble sodium alginate–gelatin complexes form as a result of strong electrostatic interactions between the positively charged amino groups of gelatin and the negatively charged carboxyl groups of sodium alginate. As the pH further decreases, a drop in the absorbance values is observed. This is due to the fact that the pH decrease causes protonation of carboxyl groups present in the sodium alginate molecule, which leads to a decrease in the total charge in the sodium alginate molecule, which leads to the fewer and weaker interactions between biopolymers [18,30].

Four types of critical values of pH (pH_c_, pH_φ1_, pH_opt_, pH_φ2_) can be graphically defined regarding the dependence of optical density on the pH. The dependence of critical pH values on Z for systems containing fish gelatin and sodium alginate is shown in Figure 7.

As can be seen in Figure 7, the values of pH_c_, pH_φ1_, pH_opt_, and pH_φ2_ depend on the mass ratio of the components in the system. As Z increases from 0.08 to 1, a drop in the critical pH values is observed. Thus, as Z increases in the SA-FG mass ratio range from 0.01 to 1.00, the pH_c_, pH_φ1_, pH_opt_, and pH_φ2_ values shift to a more acidic region: from 7.0 to 4.6, from 6.8 to 4.3, from 6.6 to 2.8, and from 6.0 to 2.7, respectively.

In the pH range from 8.0 to pH_φ1_, SA and FG co-dissolve. As the pH decreases from pH_c_ to pH_φ1_, soluble complexes are formed in SA-FG aqueous mixtures due to the weak electrostatic interactions of FG with SA. Thus, the system is a single-phase system in which individual negatively charged sodium alginate molecules and soluble SA-FG complexes are present. In the range of pH_φ1_ and pH_φ2_, the phenomenon of complex coacervation is observed. As a result of the strong electrostatic interaction of biopolymers—wherein the conditions are optimal for biopolymer interactions with neutralized charges—insoluble complexes are formed. In this case, mixtures of sodium alginate with gelatin represent a two-phase system in which one phase is represented by molecules of the dissolving agent, and the other phase is enriched by SA-FG complexes. In the range from pH_φ1_ to pH_φ2_, the SA-FG complexes visibly aggregate. Below pH_φ2_, dissociation of the alginate–sodium gelatin complexes are observed as the sodium alginate chains become increasingly protonated.

Thus, a gradual decrease in pH allows gelatin to neutralize the charge of sodium alginate because an increase in the charge of gelatin increases its binding to the polyelectrolyte. Thus, the complex charge approaches a neutral value. This contributes to increased associations and, eventually, to phase separation.

TEM and SEM images were also captured to study the effect of pH on the structure of SA-FG complexes at Z = 0.07 with a total FG concentration of 0.1% (Figure 8). TEM images revealed that the maximum number of complex particles was observed at pH = 5.5 with a minimum particle size (Figure 8a). At pH 4.5, the number of particles decreased significantly due to the dissociation of the complexes (Figure 8b).

The analysis of SEM images allows us to conclude that complex coacervates have different structures. The most homogeneous surface, characterized by the inclusion of small spherical particles (0.1–4 μm), is observed at pH = 5.5 (Figure 8e). At pH = 4.5, oval cavities ranging in size from 1 to 8 µm are observed on the surface of the coacervate phase (Figure 8d). The least homogeneous, friable structure is characteristic of the system obtained at pH = 6.0 (Figure 8f).

### 3.3. Influence of Ionic Strength and Cation Type

The presence of salts in the protein–polysaccharide system can result in decreased electrostatic interaction between protein and polysaccharide due to masking of the total charge carried by macromolecules, which leads to a decrease in the degree of complexation [48,49]. The effect of ionic strength on the interaction of gelatin and sodium alginate during acid titration was studied in aqueous solutions of sodium and calcium chlorides. The NaCl and CaCl_2_ concentration was varied from 1 to 200 mM at a mass ratio of Z = 0.07 and a gelatin concentration of 0.1% wt.

We found that the presence of electrolytes has a strong influence on the phase behavior of aqueous mixtures of sodium alginate with fish gelatin (Figure 9). As the concentration of the electrolyte increases, the absorbance decreases, and the peak position shifts to more acidic pH values. At NaCl concentrations from 1 to 10 mM, formation of the coacervate phase was observed, although the release of insoluble sodium alginate coacervates with gelatin occurred at lower pH values compared with the system containing no electrolyte (C(NaCl) = 0). At sodium chloride concentrations from 50 to 200 mM, no phase coacervation was observed, and the system is a transparent single-phase, which is indicative of almost complete suppression of the electrostatic interaction between gelatin and sodium alginate molecules because the increase in ionic strength led to an accompanying increase in the condensation of low-molecular counterions on the biopolymer surface. When NaCl was added to the SA-FG aqueous mixtures, the critical pH values (pH_c_, pH_φ1_, pH_opt_, pH_φ2_) shifted toward lower values (Figure 9a). This shift in the critical pH values is explained by the shielding effect of NaCl on sodium alginate and gelatin. CaCl_2_ caused even greater suppression of the interaction between SA and FG due to the larger cation radius (Figure 9b).

Complexation occurs only when gelatin is sufficiently protonated at lower pH values. It should be noted that the absorption maximum also decreases as the electrolyte concentration increases. Similar effects were observed for systems containing fish gelatin and sodium alginate [17], whey protein and carrageenan [50], bovine whey albumin and pectin [51], and pea protein isolate and hummiarabic [52].

## 4. Conclusions

The results of this study indicate that SA-FG aqueous mixtures exhibit different phase behaviors depending on the pH and ionic strength. During acid titration, it was shown that, as the mass ratio of the components (Z) was increased from 0.01 to 1.00, the turbidity of the system increased, and the optical density maximum shifted to the acidic region from pH 6.6 to 2.8. A decrease in pH induces complex coacervation in SA-FG aqueous mixtures. In the pH range from 8.0 to pH_φ1_, aqueous SA-FG mixtures represent a single-phase system containing single negatively charged SA molecules and soluble SA-FG complexes that form due to weak electrostatic interactions. In the pH_φ1_ to pH_φ2_ range, a biphasic system is formed in which one of the phases consists of insoluble SA-FG complexes. The maximum observed interaction between SA and FG occurs at pH_opt_, and a further decrease in pH to pH_φ2_ causes visible aggregation of SA-FG complexes. Below pH_φ2_, complex disassociation occurs as the SA chains become increasingly protonated.

With an increase in SA-FG mass ratios from 0.01 to 1.00, the boundary values of рН_c_, рH_φ1_, рH_opt_, and рH_φ2_ become more acidic: shifting from 7.0 to 4.6, from 6.8 to 4.3, from 6.6 to 2.8, and from 6.0 to 2.7, respectively.

The addition of NaCl or CaCl_2_ leads to suppression of the electrostatic -interaction between gelatin and sodium alginate molecules, whereas interaction is suppressed almost completely at high concentrations of NaCl or CaCl_2_ (100 and 200 mM), and no coacervation occurs in the SA-FG system.

Understanding the phase behavior and patterns of complex formation in aqueous mixtures of SA-FG will allow the physical and chemical properties of polyelectrolyte complexes of SA-FG to be regulated in order to obtain compositions for developing new functional materials for different applications, such as encapsulation, textural modification, and stabilization in food systems.

## Figures and Tables

**Figure 1 polymers-15-02253-f001:**
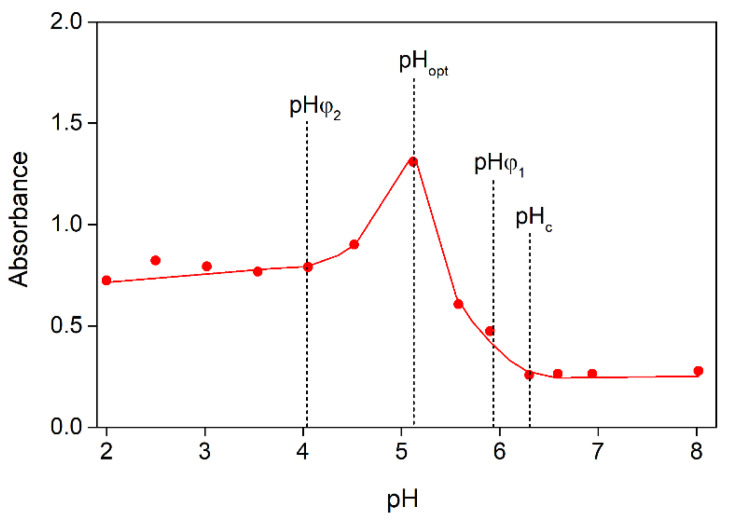
Changes in absorbance during the acid titration of aqueous SA-FG mixture prepared at Z = 0.10, where λ = 400 and T = 25 °C.

**Figure 2 polymers-15-02253-f002:**
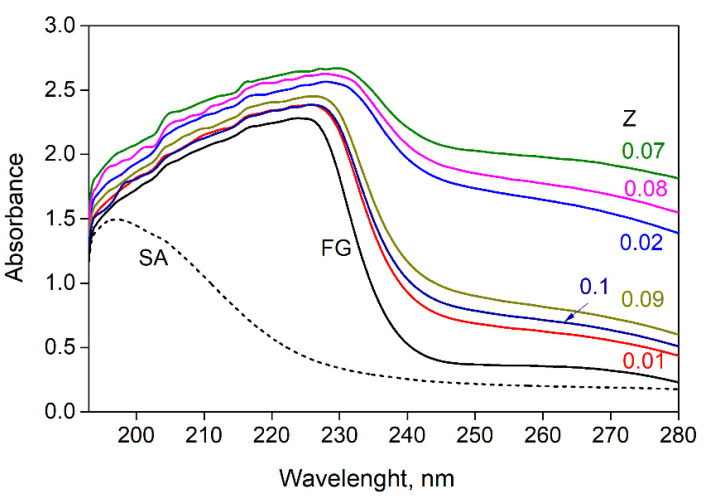
UV absorption spectra for sodium alginate (C_SA_ = 0.2 wt%), gelatin (C_FG_ = 0.2 wt.%), and SA-FG aqueous mixtures with varying mass ratios Z.

**Figure 3 polymers-15-02253-f003:**
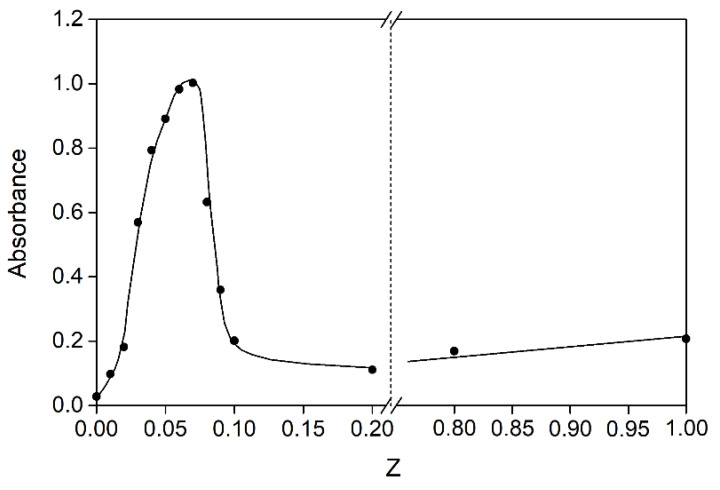
Absorbance as a function of SA-FG mass ratio in the titration of fish gelatin sol (C_FG_ = 0.2%) with SA sol (C_SA_ = 0.2%); λ = 400 nm, l = 1 cm, 25 °C.

**Figure 4 polymers-15-02253-f004:**
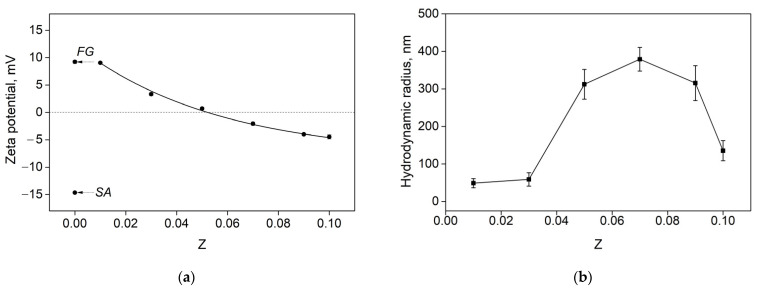
Changes in (**a**) zeta potential and (**b**) average hydrodynamic radius of particles for aqueous SA-FG mixtures prepared at different SA-FG values at 25 °C. рН ~ 6.

**Figure 5 polymers-15-02253-f005:**
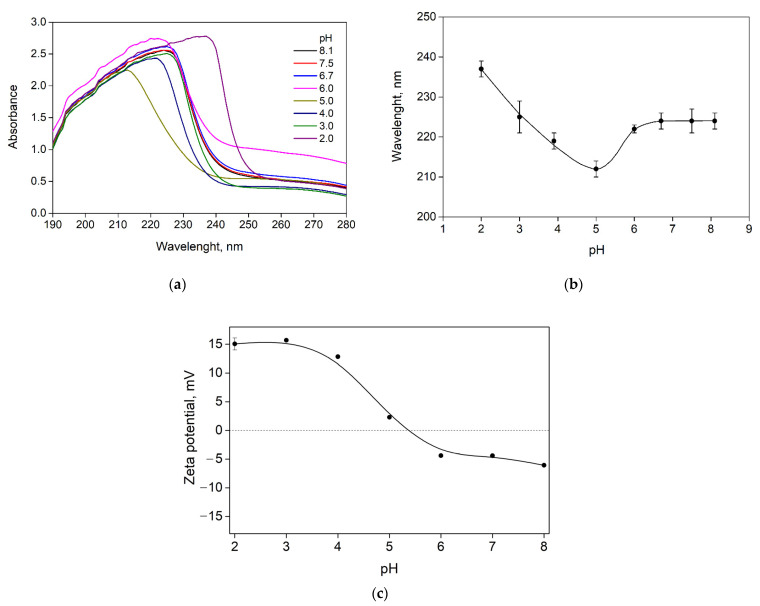
(**a**) UV absorption spectra for aqueous SA-FG mixtures prepared at different pH values and Z = 0.07; changes in (**b**) maximum absorbance and (**c**) zeta potential during acid titration of the SA-FG mixture prepared at Z = 0.07. T = 25 °C.

**Figure 6 polymers-15-02253-f006:**
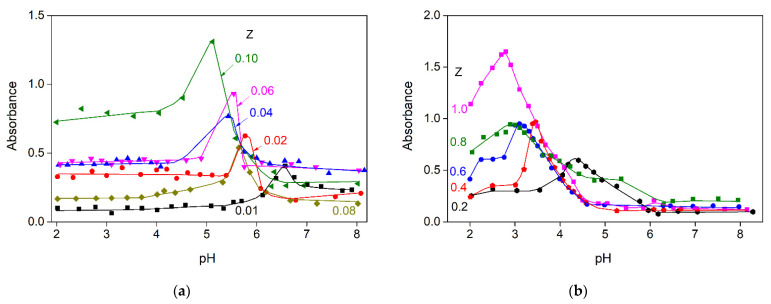
Changes in absorbance during the acid titration of aqueous SA-FG mixtures prepared at different Z values: (**a**) 0.01–0.10; (**b**) 0.1–1.0. λ = 400, T = 25 °C.

**Figure 7 polymers-15-02253-f007:**
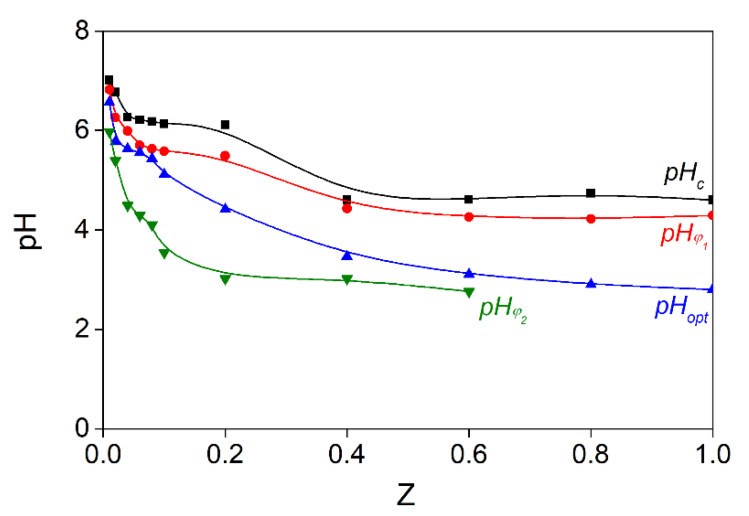
Changes in critical pH values during acid titration of aqueous SA-FG mixtures prepared at different Z values and T = 25 °C.

**Figure 8 polymers-15-02253-f008:**
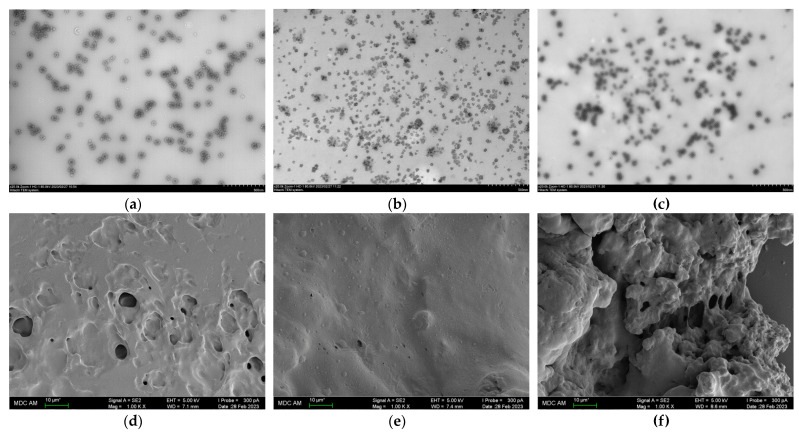
(**a**–**c**) TEM images of polymer-pure phase and (**d**–**f**) SEM images of SA-FG coacervates prepared at Z = 0.07 at different pH: (**a**,**d**) pH = 4.5; (**b**,**e**) pH = 5.5; (**c**,**f**) pH = 6.0. Z = 0.07.

**Figure 9 polymers-15-02253-f009:**
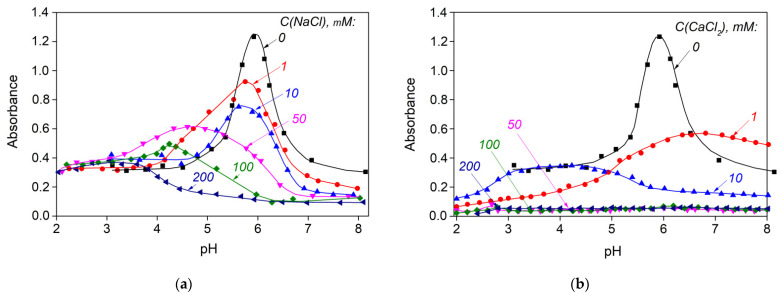
Changes in absorbance during the acid titration of aqueous SA-FG mixtures prepared at different concentrations of (**a**) NaCl and (**b**) CaCl_2_, where Z = 0.07, λ = 400, and T = 25 °C.

**Table 1 polymers-15-02253-t001:** Amino acid composition (amino acid content, g/100 g protein) of gelatin.

Amino Acids	Content, g/100 g Protein
Glycine	18.5
Proline	12.2
Hydroxyproline	7.5
Aspartic acid	5.6
Glutamic acid	9.1
Serine	6.6
Histidine	1.9
Threonine	2.7
Arginine	7.7
Alanine	9.3
Taurine	3.7
Tyrosine	1.0
Valine	2.1
Methionine	1.8
Isoleucine	1.6
Leucine	2.9
Lysine	3.5
Phenylalanine	2.3

## Data Availability

Not applicable.

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
