# Peer review of "Phase Behavior of Aqueous Mixtures of Sodium Alginate with Fish Gelatin: Effects of pH and Ionic Strength"

_polymers, 2023, doi:10.3390/polym15102253_

Round 1
Reviewer 1 Report
Following minor changes required before acceptance.
1) Line 52, attractions could be replaced with interaction.
2) Line 74, studied could be replaced with reported.
3) Quality of graph (figure 1, 3a) should be improved.
Correctly explained.
Author Response
We are grateful to the distinguished referee for useful remarks and suggestions. All the corrections made to the manuscript are marked up using the ‶Track Changes″ function.
Response to Reviewer 1 Comments
Point 1: Line 52, attractions could be replaced with interaction.
Response 1: Thank you for this comment. “Attractions” was replaced with “interactions”.
Point 2: Line 74, studied could be replaced with reported.
Response 2: “Studied” was replaced with “reported”.
Point 3: Quality of graph (figure 1, 3a) should be improved.
Responce 3: Thank you so much. According to your suggestion, we improved the quality of Figure 1 and Figure 3a to 600 dpi.
Reviewer 2 Report
The study is interesting and approaches a relevant topic. The authors obtained promising results and adequately discussed them. Overall, the manuscript has no obvious flaws, requiring a minimal revision to improve some details, as follows:
Title – It should be complemented by focusing on the main findings of the study;
Abstract – Objectives must be clearly cited. The results do not support conclusions. Affirming that the materials investigated could be applied to those functions is out of the scope of this study. Please, revise it. Keywords should be improved as well.
Introduction, material and methods, and results are appropriate and well-design. I must mention that the term solution is inadequate for describing systems with high-molecular-weight molecules, such as polymers and gelatin; it should be dispersions.
The conclusion section is too wordy and must be shortened to improve readability. Besides, the authors are recommended to avoid drawing conclusions not supported by their findings. The potential application of these materials for food and pharmaceutical purposes might be possible, but additional experiments are required to keep the hypothesis.
Author Response
We are very grateful to the reviewer for carefully reading our manuscript and for making comments and remarks, which are certainly very useful for improving the article.
All the corrections made to the manuscript are marked up using the ‶Track Changes″ function.
Response to Reviewer 2 Comments
Point 1: Title – It should be complemented by focusing on the main findings of the study.
Response 1: The title of the article has been edited slightly.
Point 2: Abstract – Objectives must be clearly cited. The results do not support conclusions. Affirming that the materials investigated could be applied to those functions is out of the scope of this study. Please, revise it. Keywords should be improved as well.
Response 2: Thank you so much. We improved abstract and keywords according to your suggestions.
Point 3: Introduction, material and methods, and results are appropriate and well-design. I must mention that the term solution is inadequate for describing systems with high-molecular-weight molecules, such as polymers and gelatin; it should be dispersions.
Response 3: Thank you for your suggestion. We replaced the word "solution" with "dispersion" where applicable.
Point 4: The conclusion section is too wordy and must be shortened to improve readability. Besides, the authors are recommended to avoid drawing conclusions not supported by their findings. The potential application of these materials for food and pharmaceutical purposes might be possible, but additional experiments are required to keep the hypothesis.
Response 4: Thank you for this comment. The conclusion section has been revised. We absolutely agree that additional experiments are needed to confirm the hypothesis, which is the goal of our future work. However, the use of such materials for food and pharmaceutical purposes has been repeatedly demonstrated in the scientific literature, due to which we allowed ourselves to make this assumption.
Reviewer 3 Report
Although the study of the influence of pH and ionic strength on protein-polysaccharides forming complexes and coacervates has been extensively reviewed, the association of fish gelatin with sodium alginate adds new data on the subjet.
Author Response
The authors are grateful to the distinguished reviewer for attention to our work.